# Multiple omics analyses and experiments validation identify PRDX3 as a biomarker of prognosis and antioncogene in kidney clear cell carcinoma

Yanyan Li[1], Xiangyuan Tan[2], Zhanzhan Li[2], Peng Chen [3]*

1 Department of Nursing, Xiangya Hospital, Central South University, Changsha, Hunan, P.R. China,
2 Department of Oncology, Xiangya Hospital, Central South University, Changsha, Hunan, P.R. China,
3 Department of Orthopedics, Xiangya Hospital, Central South University, Changsha, Hunan, P.R. China

* chenpengsurgery@hotmail.com

## Abstract

Peroxiredoxin 3 (PRDX3), a key mitochondrial redox enzyme, has been implicated in malignant tumor initiation and development. However, the its biological function and clinical relevance remain unclear. Here, we performed a systematic pan-cancer analysis of PRDX3 across 33 cancers using The Cancer Genome Atlas database, examining its expression patterns, gene alterations and mutations, methylation, sub-cellular localization, signaling pathways, tumor microenvironment, immune infiltration, and associations with clinical outcomes. We further validated the biological function role of PRDX3 in kidney clear cell carcinoma (KIRC) through reverse transcription quantitative polymerase chain reaction, Western blotting, Transwell assays, and scratch assays. PRDX3 was differentially expressed in cancers, and its expression appeared to be influenced by copy number variation and methylation status. PRDX3 levels were significantly associated with patient prognosis in multiple tumor types, suggesting context-dependent roles in tumor biology. Functional analyses indicated that PRDX3 may affect tumor progression through programs related to cell-cycle regulation, metabolism, and redox processes. In KIRC models, PRDX3 overexpression suppressed malignant phenotypes and was accompanied by changes in PPAR signaling pathway. Collectively, our results support PRDX3 as potential prognostic biomarker and suggest a tumor-suppressive role in KIRC, although further mechanistic and in vivo validation is warranted.

## Introduction

Malignant tumors remain a leading cause of death worldwide [1]. In a World Health Organization (WHO) assessment of 183 countries surveyed, cancer ranks first or second among causes of death before the age 70 years in 112 countries and third or fourth in 23 other countries. In 2020, the international Agency for Research on

**Data availability statement:** All relevant data are within the manuscript and its Supporting Information files.

**Funding:** The author(s) received no specific funding for this work.

**Competing interests:** The authors have declared that no competing interests exist.

Cancer reported approximately 10 million cancer-related deaths globally. In terms of incidence, breast cancer accounted for 2.3 million new cases (11.7%), followed by lung cancer (11.4%), colorectal cancer (10.0%), prostate cancer (7.3%), and stomach cancer (5.6%) [2]. Lung cancer caused 1.8 million deaths annually (18%). While colorectal cancer was the second leading cause of cancer mortality (9.4%), followed by liver cancer (8.3%), stomach cancer (7.7%), and female breast cancer (6.9%) [3]. Kidney cancer also represents a substantial global burden. In 2022, GLOBOCAN estimated 434,840 new cases and 155,953 deaths worldwide, ranking 14th for incidence and 16th for mortality, with a 5-year prevalence globally. Despite advances in management, outcomes remain stage-dependent [4]. Additionally, the WHO estimates a 60% increase in global cancer cases over the next 20 years [5]. Therefore, clarifying tumor etiology and identifying clinically useful biomarkers and therapeutic targets remain critical for earlier detection, improved risk stratification, and better long-term outcomes. Importantly, biomarker research is not only about statistical association but also about eventual clinical utility. A biomarker is broadly defined as a measurable characteristic that reflects normal or pathogenic processes or responses to an intervention, and cancer biomarkers may serve diagnostic, prognostic, predictive, or monitoring roles depending on their intended "context of use" [6]. In general, discovery-stage candidates should be evaluated along a translational continuum that includes analytical validation (reliable and reproducible measurement), clinical validation (replication of associations with relevant clinical outcomes), and finally clinical utility (demonstrated benefit for prognosis, stratification, or treatment decision-making). Within this framework, pan-cancer analyses can be particularly informative at the discovery stage by assessing the robustness and tumor-type specificity of biomarker signals and by prioritizing cancer types for focused validation and mechanistic studies.

Peroxiredoxins (PRDXs) are a family of thiol-dependent peroxidases in mammalian cells comprising at least six isoforms (PRDX1–6). PRDX1–2 and PRDX5–6 are primarily localized in the cytoplasm, PRDX4 is mainly found in the endoplasmic reticulum, and PRDX5 can also localize to peroxisomes and mitochondria [7]. PRDX3 is predominantly localized in mitochondria and functions as a major $H_2O_2$-scavenging enzyme [8]. Under oxidative stress, PRDX proteins can undergo redox-dependent modifications and may participate in stress signaling, including forming stable oxidative dimers that can be secreted extracellularly [9,10]. PRDX3 is considered one of the most abundant and efficient mitochondrial $H_2O_2$-removing enzymes and can eliminate a large proportion of matrix $H_2O_2$ through redox cycling, thereby limiting oxidative stress and apoptosis and reducing cellular damage [11]. Consistently, PRDX3 overexpression in transgenic models reduces mitochondrial $H_2O_2$ production and oxidative injury compared with controls [11]. In the craniocerebral trauma model, the target gene can reduce cell damage by protecting mitochondrial function. Moreover, transgenic mice with PRDX3 overexpression can reduce mitochondrial $H_2O_2$ production and thus reduce oxidative damage compared with normal mice [12]. PRDX3 has also been implicated in ferroptosis: mitochondrial lipid peroxides can trigger excessive PRDX3 oxidation, leading to post-translational modifications of its catalytic

cysteine; oxidized PRDX3 may translocate and contribute to ferroptotic processes by impairing cystine uptake [13,14]. Beyond redox regulation, PRDX3 has been linked to tumor-related phenotypes, including proliferation, migration, metastasis, and drug response. For example, PRDX3 overexpression suppresses proliferation in thymoma cells by reducing intracellular $H_2O_2$, whereas in hepatocellular carcinoma cells PRDX3 has been reported to influence invasion and extracellular matrix remodeling [15]. Loss of PRDX3 has been associated with poor survival and metastasis in uveal melanoma [16], and PRDX3 contributes to the resistance of gastric cancer cells against cisplatin [17]. In non-small cell lung cancer, resistance to osimertinib has been linked to altered PRDX3 SUMOylation regulated by ubiquitin-specific peptidase 7 [18].

In the present study, we performed a comprehensive pan-cancer analysis of PRDX3 in The Cancer Genome Atlas (TCGA), evaluating its expression patterns, genomic variations, protein–protein interaction networks, subcellular localization, and associations with patient prognosis. We further explored potential mechanisms through functional enrichment and immune infiltration analyses.. Based on these results, we prioritized kidney renal clear cell carcinoma (KIRC) for downstream validation and assessed the prognostic relevance of PRDX3 in KIRC. Finally, we conducted in vitro phenotypic experiments to preliminarily validate the effects of PRDX3 on KIRC cells. Collectively, our findings suggest that PRDX3 may serve as a candidate biomarker for prognosis and stratification in specific tumor contexts and provide a foundation for future mechanistic and translational studies..

## Materials and methods

### Patients and samples

We downloaded PRDX3 gene expression profiling, gene alterations, copy number variation, and clinical data of 33 types of cancers from TCGA (n = 10977) (https://portal.gdc.cancer.gov/) and extracted the normal tissue of PRDX3 expression profiling (n = 730), on November 24, 2024. Protein data were extracted from the human protein atlas (https://www.protein-atlas.org/). All abbreviations were listed in S1 Table. Ethical approval and informed consent were not required because all data were obtained from publicly accessible, de-identified datasets, and no individually identifiable information was used or generated in this study.

### Bioinformatic analysis

We adopted "survival" and "survminer" packages for prognosis analyses. Four clinical outcomes were applied, including the following outcomes: overall survival (OS), progression-free interval (PFI), disease-specific survival (DSS), and disease-free interval (DFI). Kaplan-Meier and univariate cox regressions were performed to evaluate the associations between PRDX3 expression and clinical prognosis [19].

Gene set variation analysis (GSVA) was conducted using "GSVA" packages. Samples were dichotomized into high- and low-PRDX3 groups according to the median PRDX3 expression, and pathway activity differences between groups were compared. Hallmark gene sets were used as the primary reference. In addition, curated gene sets related to regulated cell death pathways from prior studies were included, covering ferroptosis, autophagy, apoptosis, necroptosis, pyroptosis, endocytosis, and cuproptosis [20–22].

Genetic alterations and protein mutation frequencies of PRDX3 across tumor types were analyzed using cBioPortal (http://www.cbioportal.org)[21]. Protein–protein interaction analysis was performed using the comPPI database (https://comppi.linkgroup.hu/protein_search/interactors/P30048) [22]. Tumor mutation burden (TMB) and microsatellite instability (MSI) were also analyzed.

The immuno-gene module of TIMER2.0 (http://timer.cistrome.org/) was used to evaluate associations between PRDX3 expression and immune infiltration in TCGA. The immune infiltration was estimated based on TIMER, CIBERSORT, QUANTISEQ, XCELL, EPIC, MCPCOUNTER and TIDE algorithms [20]. P-values and partial correlation values were obtained through purity correction Spearman's rank test. Heat maps were used to visualize the data. We further examined

correlations between PRDX3 expression and immune-related genes, chemokines, and chemokine receptor genes across cancers.

## Cells, cell culture and transfection

Cell lines 786−0 and Caki-1 were procured from Procell (Wuhan, China) and cultured in Dulbecco's modified Eagle's medium supplemented with 10% fetal bovine serum (Gibco, USA) in a 37 °C incubator with 5% $CO_2$. Stable PRDX3-overexpressing cell lines were generated by lentiviral transduction.. The PRDX3 gene was successfully integrated into the cell genome, enabling its expression. Flow cytometry was employed to sort GFP-positive cells, then monoclonal cells were selected and inoculated into 96-well plates. Fluorescence microscopy was conducted for two weeks, and subsequently, positive clones were transferred to 6-well plates to facilitate expanded culture. Stable PRDX3-overexpressing clones were selected based on PRDX3 expression levels. Cells transduced with the empty vector served as controls. For clarity, the PRDX3-overexpression and control groups are referred to as PRDX3(+) and PRDX3(−), respectively..

## Reverse-transcription quantitative polymerase chain reaction (RT-qPCR)

Total RNA was reverse-transcribed using the RevertAid First Strand cDNA Synthesis Kit according to the manufacturer's instructions. Quantitative PCR was performed using SYBR Green on a CFX96 Real-Time PCR System (Bio-Rad). GAPDH was used as the internal reference. Primer sequences are listed in S2 Table. Experiments were performed in triplicate. Relative mRNA expression was calculated using the ΔCt method: $\Delta Ct = Ct(target) - Ct(GAPDH)$ [23].

## Western blot

Total protein was extracted and quantified, and 50 μg of protein per sample was separated by SDS–PAGE using a 10% resolving gel and 5% stacking gel (100 V, ~90 min). Then a 45 pm polyvinylidene fluoride membrane was subjected to 300 mA ice bath transfer for 90 minutes. After the transfer, the membrane was sealed with 5% skimmed milk powder and rinsed with tris-buffered saline and 0.05% Tween-20 (TBST) for 30 minutes (5 minutes × 3). The primary antibodies (Anti-PPAR alpha: ab233078; PRDX3: ab222807) were applied in accordance with the product description, followed by an overnight incubation at 4 °C. After a TBST rinse (5 minutes × 3), secondary antibodies were added and left to incubate at room temperature. Another TBST rinse (5 minutes × 3) ensued. Equal parts of liquid A and liquid B in enhanced chemiluminescence (ECL) were mixed, and ECL luminescent liquid was added for target strip image detection using a multifunctional imaging system. GAPDH was used as the loading control.

## Cell migration, invasion, and scratch assays

Migration and invasion assays were performed using Transwell chambers in 24-well plates. Briefly, 0.5 mL complete medium was added to the lower chamber, and 200 μL cell suspension was added to the upper chamber. Cells were incubated at 37 °C with 5% $CO_2$ for 30 h. Chambers were then fixed in paraformaldehyde for 30 min, stained with crystal violet for 30 min, washed with water, and non-migrated cells on the upper surface were removed using a cotton swab. Images were captured under an inverted microscope, and cells were counted in representative fields..

Under an inverted microscope, images were captured, and data were recorded from the central area of each well. The cells that crossed the microporous membrane within the same field of view were then counted. The transfected cells of each group were allowed to grow normally in 6-well plates with a cell density of 70%−80% in a complete serum-containing medium. Once the cells became overgrown, the medium was removed using a 100 ul gun tip perpendicular to the bottom of each well longitudinally, with three strokes in each hole. After gentle washing twice with phosphate buffer solution (PBS), the cells were cultivated in a serum-free RPMI 1640 medium for 0 hours. Pictures were taken and recorded with an optical microscope at 0 h and 36 h. The scratch test was filmed, and the relative healing area of each cell was calculated.

Cells with a larger relative healing area demonstrated a stronger migration ability. The experiments were performed three times.

## Statistical analysis

Data are presented as mean ± standard deviation (SD). Comparisons between two groups were performed using a two-tailed independent-samples t test. Receiver operating characteristic (ROC) curve analysis was used to evaluate the diagnostic performance of PRDX3 in KIRC. A two-sided P value < 0.05 was considered statistically significant. All plots were generated using GraphPad Prism 9.0..

## Results

### General characteristics of PRDX3

Data from the human protein Atlas indicated that PRDX3 had low tissue specificity and is broadly expressed across most human tissues, with the highest expression observed in the liver, tongue, kidney, skeletal muscle, and adrenal gland (Fig 1A). Subcellular localization analysis showed that PRDX3 is mainly distributed in the endosome, mitochondria, cytosol, nucleus, and plasma membrane (Fig 1B). Protein–protein interaction analysis identified 119 interacting partners, with the top 10 (ranked by interaction score) being LRRK2, HSPD1, OPA1, PARK7, DLD, MAP1LC3B, SNCA, DHRS4, RPS6KC1, and UCHL5 (Fig 1C). Data from TCGA showed that PRDX3 was significantly downregulated in cholangiocarcinoma (CHOL), kidney chromophobe (KICH), KIRC, kidney papillary cell carcinoma (KIRP), sarcoma (SARC), pheochromocytoma & paraganglioma (PCPG), and thyroid carcinoma (THCA). The expression levels of PRDX3 were significantly upregulated in colon adenocarcinoma (COAD), esophageal carcinoma (ESCA), glioblastoma multiforme (GBM), lung adenocarcinoma (LUAD), lung squamous cell carcinoma (LUSC), prostate adenocarcinoma (PRAD), stomach adenocarcinoma (STAD), and uterine corpus endometrioid carcinoma (UCEC) (Fig 1D). PRDX3 expression also varied by alteration type, suggesting that genomic events may influence its transcriptional level (Fig 1E). Copy-number alterations showed tumor-type–specific patterns, with amplification being more frequent in miscellaneous neuroepithelial tumors and deep deletions enriched in glioma (Fig 1F). In addition, PRDX3 DNA methylation was negatively correlated with its mRNA expression (S1 Fig).

### PRDX3 and clinical prognosis in cancers

We evaluated the correlations of PRDX3 with four clinical prognosis outcomes: OS, PFI, DSS, and DFI using Kaplan-Merie and cox regression (Fig 2A). For OS, PRDX3 was identified as a risk factor in CHOL, head & neck squamous cell carcinoma (HNSC), KICH, acute myeloid leukemia (LAML), pancreatic adenocarcinoma (PAAD), PRAD, and rectum adenocarcinoma (READ), whereas it was a protective factor in adrenocortical cancer (ACC), COAD, KIRC, KIRP, brain lower grade glioma (LGG), liver hepatocellular carcinoma (LIHC), mesothelioma (MESO), ovarian serous cystadenocarcinoma (OV), SARC, and skin cutaneous melanoma (SKCM) (Fig 2B). For PFI and DSS, K-M analyses and cox regression also identified a risky prognosis role in cervical & endocervical cancer (CESC), Claudin-low breast cancer (CLBC), ESCA, HNSC, KICH, LUAD, PRAD, and READ and a protective role in COAD, GBM, KIRC, LGG, LIHC, LUSC, MESO, SARC, and SKCM (Figs 2C and 2D). For DFI, directionally inconsistent results were observed for some tumor types (e.g., ACC, KIRC, and OV as risk factors; PRAD and READ as protective factors; Fig 2E), likely reflecting limited relapse-related follow-up events in certain cohorts. Overall, PRDX3 exhibited context-dependent, dual prognostic roles across cancers.

### Association between PRDX3 and pathways in cancers

We next explored the pathway differences between high- and low-expressed PRDX3 groups in cancers. The GSVA results demonstrated that PRDX3 was positively enriched in protein secretion, peroxisome, oxidative phosphorylation, mTORc1

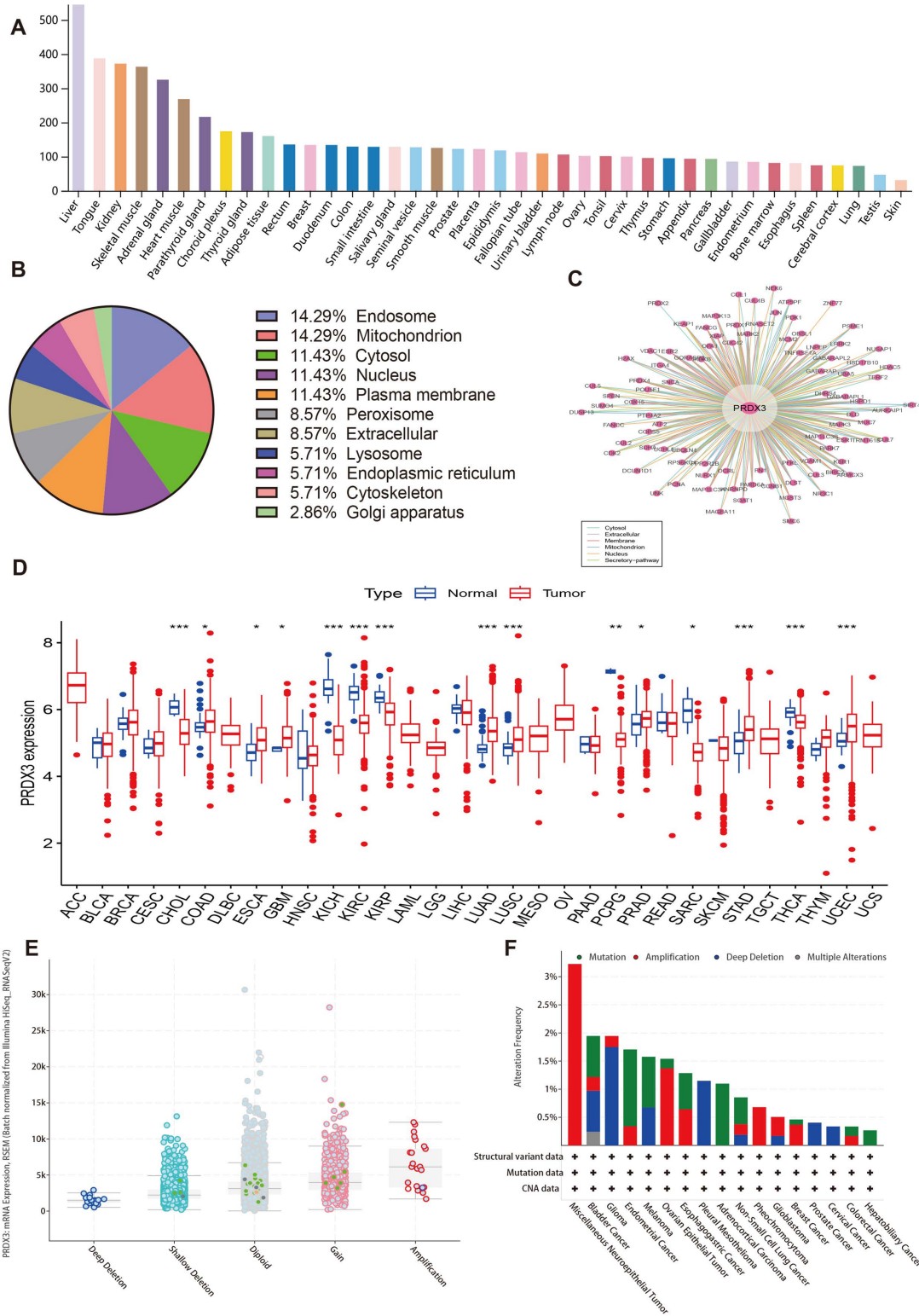

**Fig 1. Landscape of PRDX3. A:** PRDX3 expression in human organs. **B:** Sub-cellular localization of PRDX3. **C:** First-neighbor network including 119 nodes. **D:** Expression profile of PRDX3 between tumor and normal samples. **E:** PRDX3 mRNA expression and gene alteration. **F:** PRDX3 alteration frequency in different cancers.

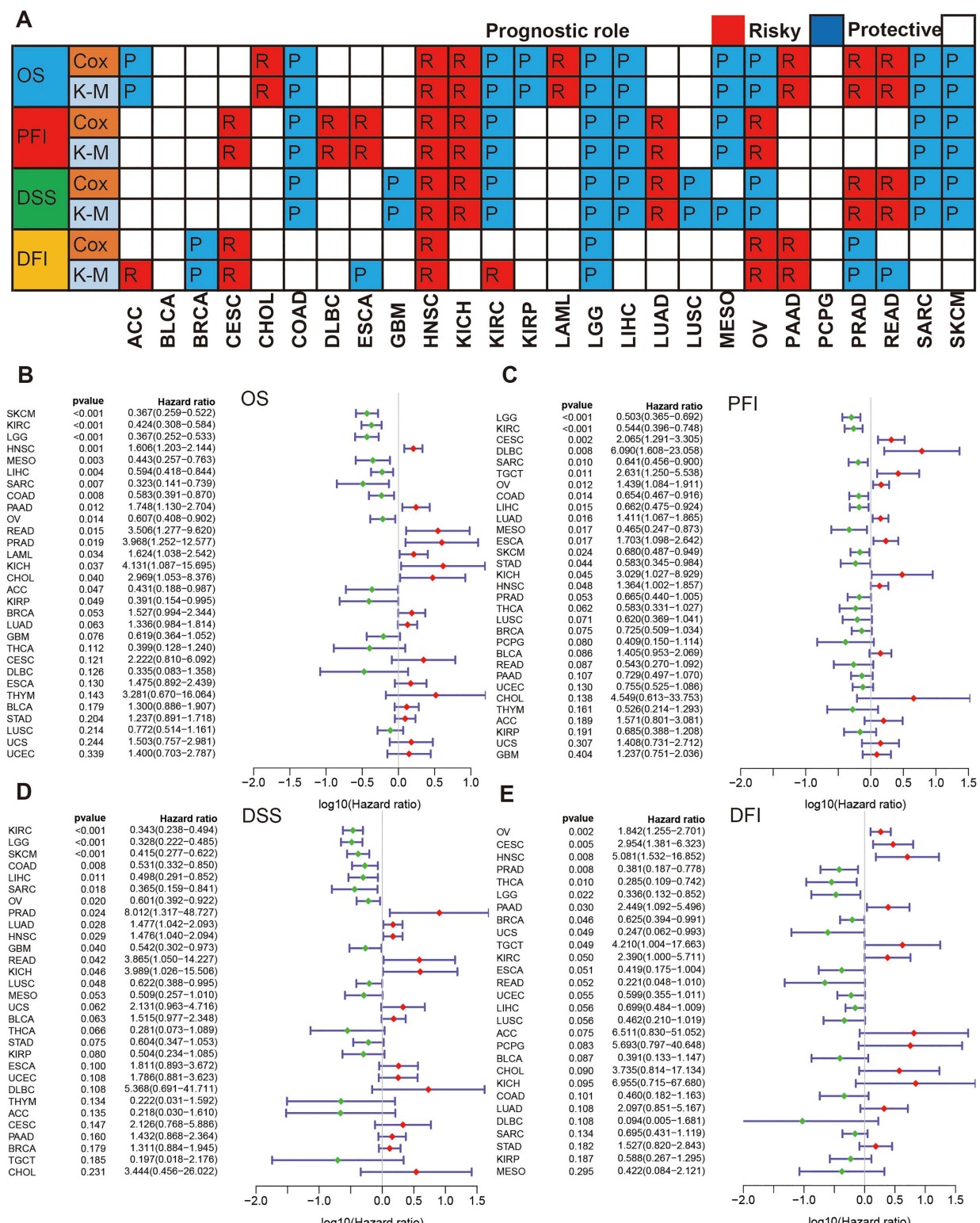

**Fig 2. PRDX3 and clinical prognosis in cancers. A:** Summary of correlations of PRDX3 with clinical prognosis using Kaplan-Meier and cox regressions (OS: overall survival; PFI: progression-free internal; DSS: disease specific survival; DFI: disease-free interval). The red R means risky, the blue P means protective, and the blank means no significance. **B-E:** Forest plot about the prognostic roles of PRDX3 in cancers: OS, PFI, DSS and DFI. The $log_{10}$ of hazard ratio (HR) is achieved. The red means the HR is >1, and the green means the HR is <1. There are no significances for these lines including zero.

signaling, fatty acid metabolism, bile acid metabolism, and androgen response, and adipogenesis in cancers (Fig 3A). In contrast, p53 pathway activity, myogenesis, KRAS signaling (down), epithelial–mesenchymal transition (EMT), apical surface, and apical junction were generally negatively enriched across cancers. Several immune- and inflammation-related programs—including interferon-α/γ responses, inflammatory response, IL6/JAK/STAT3 signaling, IL2/STAT5 signaling, hypoxia, complement, apoptosis, and allograft rejection—were significantly negatively enriched in multiple tumor types (e.g., ACC, BLCA, BRCA, CESC, CHOL, KIRC, KIRP, LIHC, LUAD, OV, PAAD, PCPG, PRAD, TGCT, THCA, UCEC, and UCS), whereas in GBM most hallmark pathways were positively associated with PRDX3 expression.

A recent study reported that PRDX3 may be an important biomarker of ferroptosis [13]. Hence, we further explored the associations between RPDX3 and some cell death-related signaling pathways. PRDX3 was positively associated with ferroptosis and cuproptosis in almost all cancers (Fig 3B). PRDX3 may induce the inactivity of apoptosis modulation signaling, necrosis pathway, autophagy, pyroptosis, endocytosis, and apoptosis in ACC, BLCA, BRCA, CESC, CHOL, LIHC, LUAD, OV, PAAD, PCPG, PRAD, TGCT, THCA, and UCS. PRDX3 can activate necrosis, pyroptosis, apoptosis, and autophagy in some cancers, such as GBM, COAD, LAML, LUSC, READ, and STAD. In KIRC, PRDX3 mainly affected ferroptosis and cuproptosis.

## Associations between PRDX3 and tumor microenvironment in cancers

We first evaluated the correlations of PRDX3 with TMB and MSI in cancers. We found PRDX3 was associated positively with TMB in CHOL, COAD, LUAD, LUSC, PRAD, READ, SKCM, STAD, TGCT, UCEC, and UCS, but negatively with BRCA and LGG (Fig 4A). Meanwhile, PRDX3 was positively associated with MSI in ESCA, KIRC, STAD, TGCT, and UCEC, but negatively with BLCA, BRCA, diffuse large B-cell lymphoma (DLBC), LAML, LUAD, and LUSC (Fig 4B). Because TMB and MSI are commonly used surrogates of neoantigen load and immunogenicity, these tumor-type–dependent directions suggest that PRDX3 may track distinct mutational/immunogenic contexts rather than a uniform "high-mutation" phenotype [24]. We further evaluated the associations of PRDX3 with immune score and stromal score about its effect on tumor microenvironment in cancers. Results showed PRDX3 was negatively associated with immune score in BLCA, BRCA, CESC, CHOL, KIRC, KIRP, LGG, LIHC, LUAD, PRAD, SARC, TGCT, THCA, UCEC except GBM and LUSC (Figs 4C-4R). Similar results were found between PRDX3 and stromal score in these cancers except GBM (S2 Fig). A negative association with immune/stromal scores can be consistent with (i) relatively higher tumor purity (fewer non-tumor immune/stromal components) and/or (ii) an immune-excluded or immune-low microenvironment in certain tumors. Clinically, microenvironmental states are often linked to reduced immune cell infiltration and may contribute to diminished benefit from immunotherapies in some contexts, supporting PRDX3 as a candidate marker for microenvironmental stratification rather than a standalone predictor.

## Association between PRDX3 and immune infiltration levels in cancers

We next examined immune infiltration signatures (Fig 5). Across many tumor types, PRDX3 tended to show negative correlations with memory B cells, class-switched memory B cells, activated myeloid dendritic cells, and endothelial cell signatures, whereas positive associations were observed for naïve B cells, plasma B cells, total B cells, and common lymphoid progenitors in several cancers. In addition, PRDX3 showed predominantly negative correlations with macrophage M0/M1, monocytes, NK cells, CD4 + effector memory T cells, Th1 cells, and regulatory T cells, while positive correlations were observed for mast cells, neutrophils, CD4 + memory (resting) T cells, Th2 cells, total CD4 + T cells, and CD8 + T cells in many cancers. Overall, The co-existence of negative and positive associations across immune subsets suggests that PRDX3 may relate to immune state remodeling rather than simply "more" or "less" immunity.

We analyzed immune-related genes. PRDX3 was significantly and negatively correlated with several immune-activation genes (MIGD2 and multiple TNFRSF members) in most cancers, with notable exceptions such as UVM, where these associations were positive (Fig 6A). In UVM, DLBC, and THYM, PRDX3 was positively correlated with multiple immune-activation

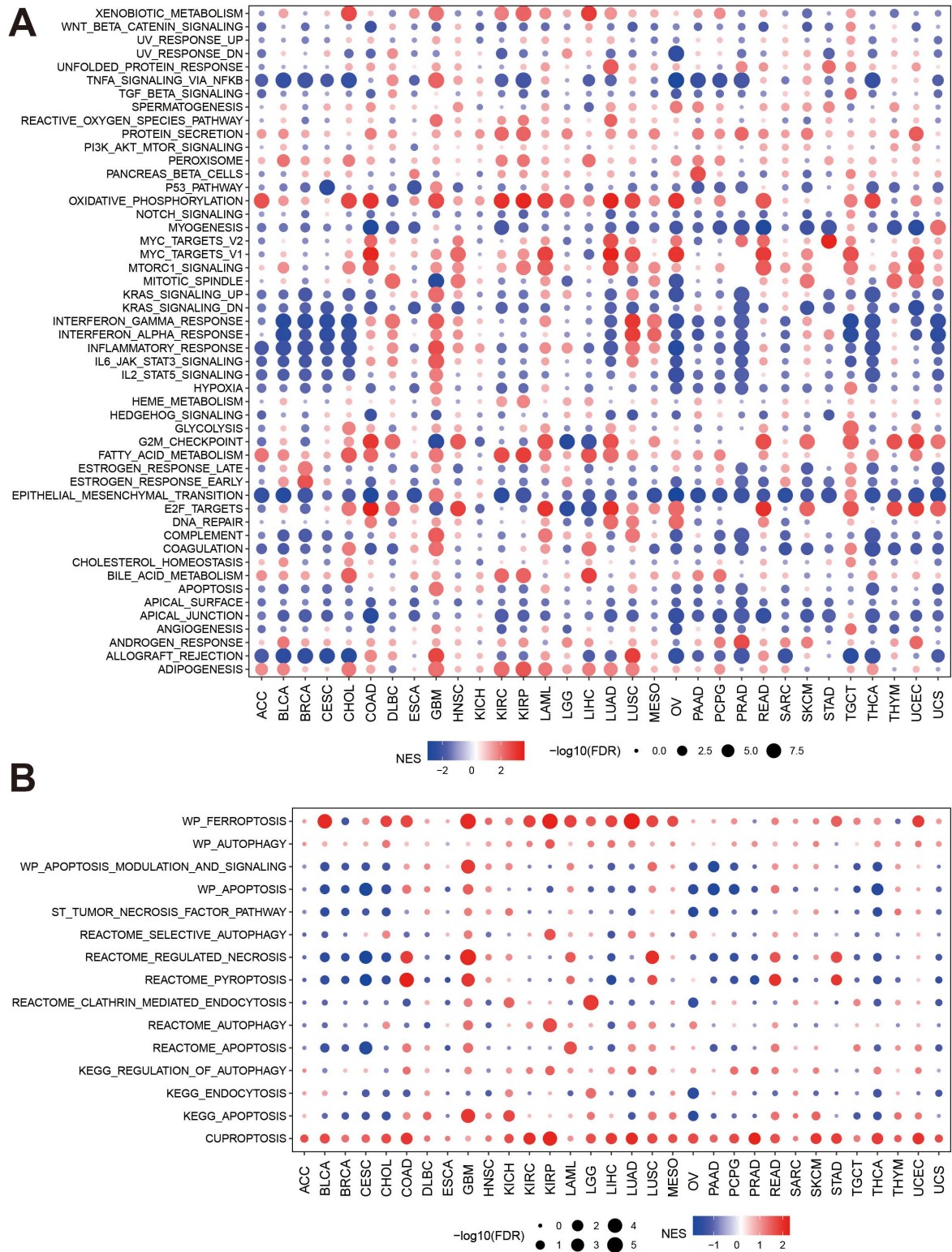

**Fig 3. PRDX3 and pathways in cancers. A:** Associations between PRDX3 and signaling pathways in cancers. **B:** PRDX3 and cell death relevant pathways in cancers. The colors of circles represent correlation, and the size of circle means -log10(FDR). The bigger the circle is, the smaller P value is. FDR: false discovery rates, and all P values were corrected using FDR. The while circle means no significance.

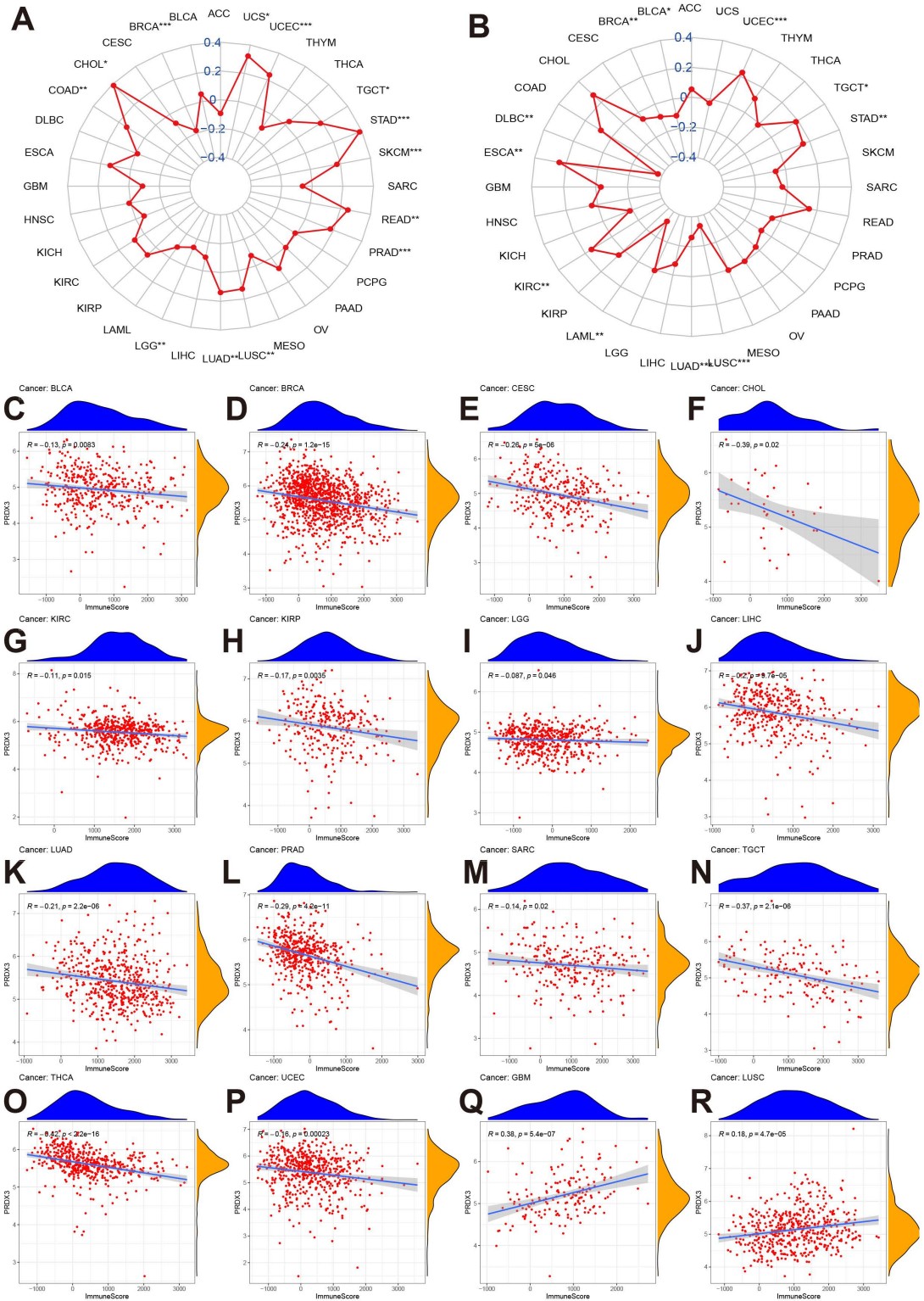

**Fig 4. PRDX3 and tumor microenvironment in cancers. A:** PRDX3 and TMB in cancers. The dots mean the correlations between PRDX3 expression and TMB in different cancers. The value>0 means positive correlation, and value<0 means negative correlation. *: P<0.05; **: P<0.01; ***: P<0.001;

****=P<0.0001. **B:** PRDX3 and MSI in cancers. The value>0 means positive correlation, and value<0 means negative correlation. *: P<0.05; **: P<0.01; ***: P<0.001; ****=P<0.0001. **C-R:** correlations of PRDX3 expression with immune scores in BLCA, BRCA, CESC, CHOL, KIRC, KIRP, LGG, LIHC, LUAD, PRAD, SARC, TGCT, THCA, UCES, UCEC, GBM, and LUSC. The x-axis=immune score in cancers, and the y-axis=PRDX3 expression cancers.

genes, including TNFSF13B, CD28/CD80/CD86, ENTPD1, IL2RA, and NT5E, among others. Tumor-type–specific patterns were also observed: PRDX3 tended to be positively associated with immune-activation genes in HNSC (with a few exceptions), but negatively associated with PCPG, KIRP, PRAD, LUAD, CESC, SARC, and ACC. In KIRC, most immune-activation genes were positively correlated with PRDX3, except for IL6R, HHLA2, and CXCL12. We also evaluated the correlation of PRDX3 with immune-inhibited genes in cancers. Immune-inhibited genes showed positive associations with PRDX3 in DLBC, UVM, and HNSC. Almost immune-inhibited genes were negatively associated with PRDX3 in TGCT, CHOL, THCA, UCS, LIHC, PCPG, PRAD, LGG, BRCA, SARC, MESO, BLCA, CESC, LAML, LUAD, OV, ACC except KDR, CD160, and ADORA2A. KDR was positively associated with PRDX3 in most cancers except LUAD and OV. CD274 was positively associated with PRDX3 in KIRP, DLBC, UVM, HNSC, SKCM, PAAD, and KIRC (Fig 6B). These patterns support a tumor-type–dependent immune phenotype. In tumors where PRDX3 correlates positively with PD-L1 and other inhibitory checkpoints, PRDX3-high cases may reflect an "inflamed but checkpoint-restrained" microenvironment (a form of adaptive immune resistance), which is mechanistically aligned with established checkpoint biology and could have implications for sensitivity to checkpoint blockade. Conversely, broadly negative correlations with immune-activation genes in other tumors are more compatible with an immune-low or excluded context, potentially linked to primary resistance.

Chemokines and chemokine receptors regulate leukocyte trafficking within tumors [25]. We observed that most chemokines were positively correlated with PRDX3 in GBM, HNSC, ESCA, THYM, STAD, COAD/READ, LUSC, PAAD, and UCEC. In contrast, chemokine genes were largely negatively correlated with PRDX3 in THCA, except for CCL15, CCL14, and CCL13. In KIRC, PRDX3 showed positive correlations with several chemokines (CXCL9, CXCL11, CXCL16, CX3CL1, CCL2, CCL22, CCL24, and CCL28) and negative correlations with others (Fig 7A). For chemokine receptors, PRDX3 exhibited positive correlations in THYM, DLBC, UVM, ESCA, and HNSC but negative correlations in THCA, CHOL, TGCT, STAD, LUAD, and LAML. In KIRC, PRDX3 was positively correlated with CCR2, CXCR1, and CXCR2, but negatively correlated with several other receptors (Fig 7B).. Because chemokine programs shape immune cell recruitment and spatial distribution, PRDX3-linked chemokine signatures may indicate differences in immune cell trafficking and thereby influence immunotherapy responsiveness.

## PRDX3 independently predicted prognosis of KIRC patients

We compared PRDX3 expression between KIRC tumors and normal tissues and found that PRDX3 was significantly downregulated in tumor samples (Fig 8A). ROC analysis showed strong diagnostic performance (AUC=0.947, 95% CI: 0.926–0.963; Fig 8B), with sensitivity and specificity of 0.879 and 0.944, respectively. Immunohistochemistry further supported low PRDX3 protein expression in KIRC (Fig 8C). PRDX3 expression decreased with advancing clinical stage (Fig 8D). In Cox regression analyses, PRDX3 was an independent prognostic factor in both univariable analysis (HR=0.661, 95% CI: 0.524–0.833, P<0.001; Fig 8E) and multivariable analysis (HR=0.625, 95% CI: 0.488–0.800, P<0.001; Fig 8F). We further constructed a risk assessment model incorporating PRDX3 and clinical parameters (Fig 8G), which showed good agreement in predicting 5-year OS based on calibration analysis (Fig 8H).

## PRDX3 inhibited KIRC progression via PPAR signaling pathway

Across our pan-cancer analyses, KIRC consistently showed: (i) significant PRDX3 dysregulation in tumor vs. normal tissues, (ii) robust clinicopathological associations, (iii) diagnostic potential in ROC-based evaluation, and (iv) consistent

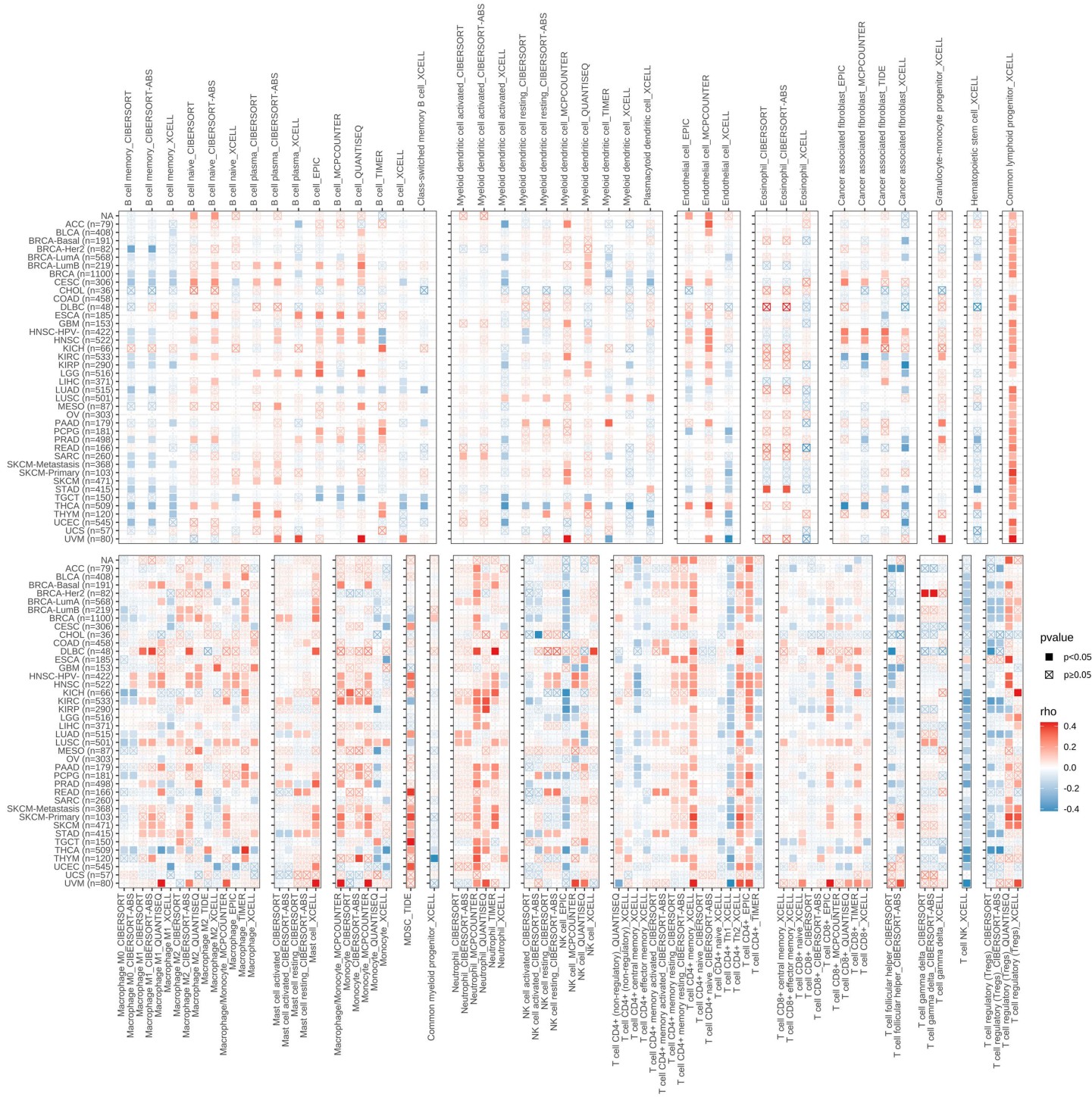

**Fig 5. Associations between PRDX3 and immune infiltration levels in cancers.** The colors of square means p values, the red is positive correlated, the blue is negative correlated, and the x square means no significance.

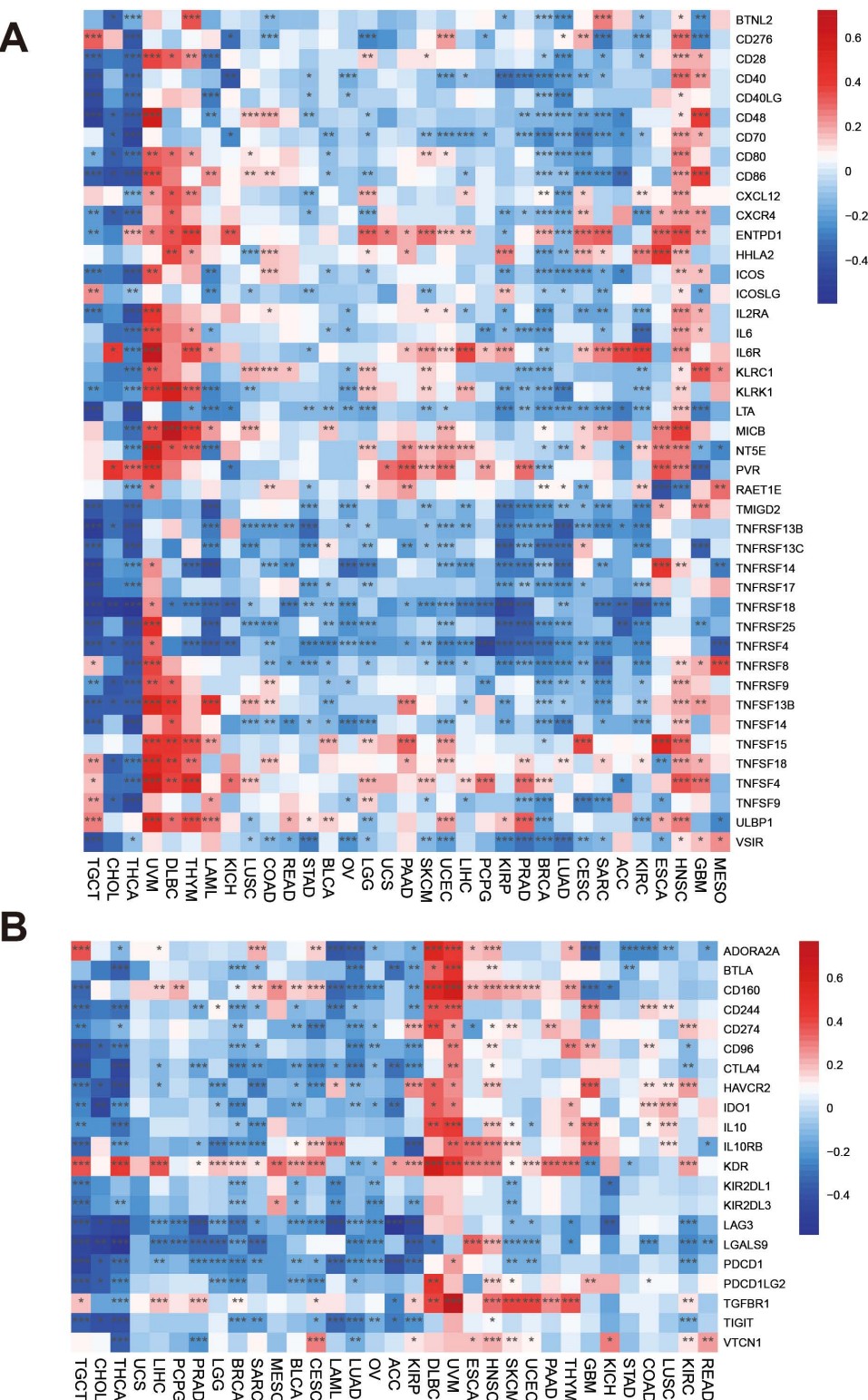

**Fig 6. Associations between PRDX3 and immune regulators in cancers. A:** PRDX3 and immune activated genes. **B:** PRDX3 and immune inhibited genes. *: $P < 0.05$; **: $P < 0.01$; ***: $P < 0.001$; ****$= P < 0.0001$.

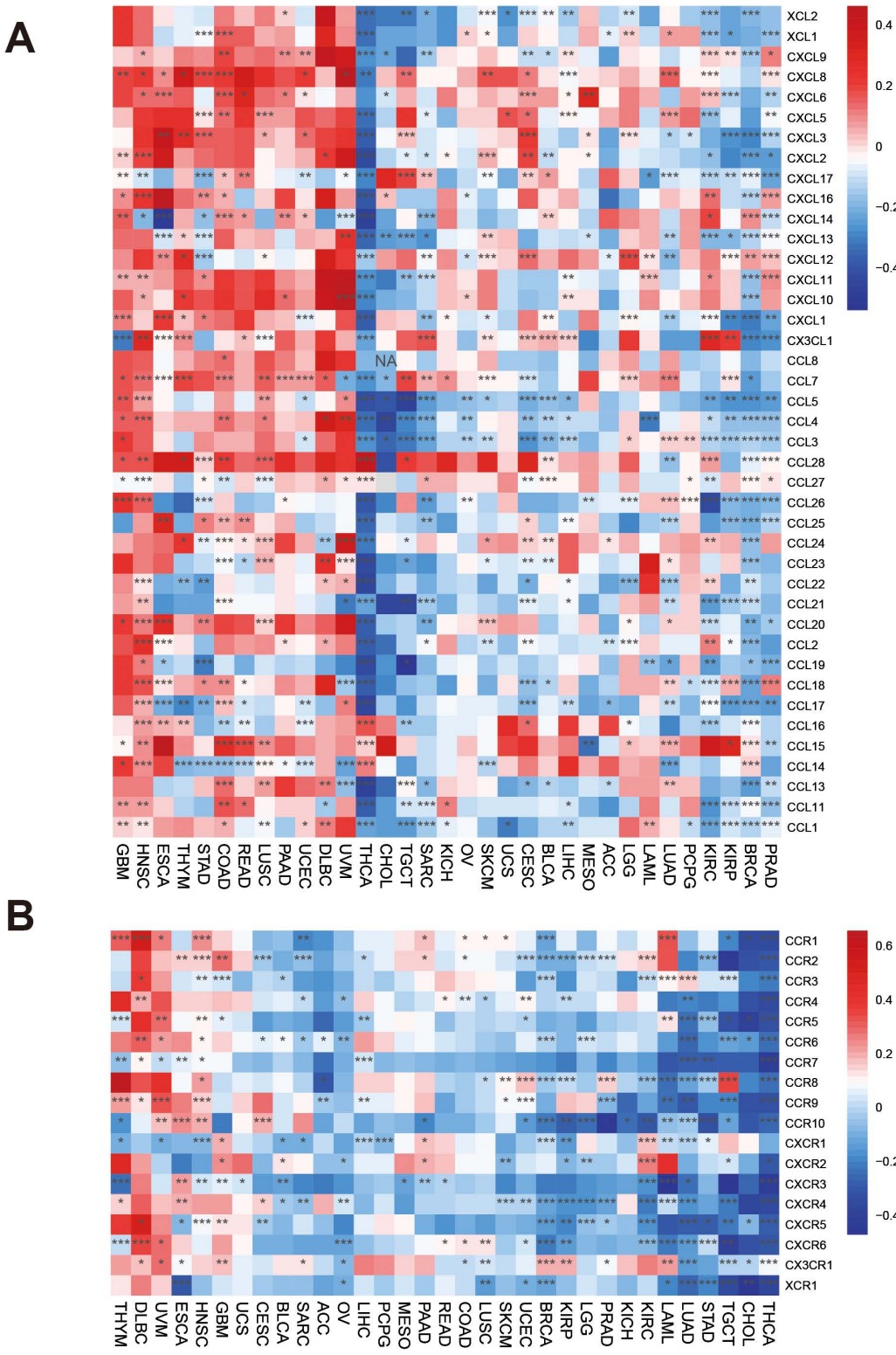

**Fig 7. Associations between PRDX3 and chemokine regulators. A:** PRDX3 and chemokine genes. **B:** PRDX3 and chemokine receptor genes. *: $P < 0.05$; **: $P < 0.01$; ***: $P < 0.001$; **** = $P < 0.0001$.

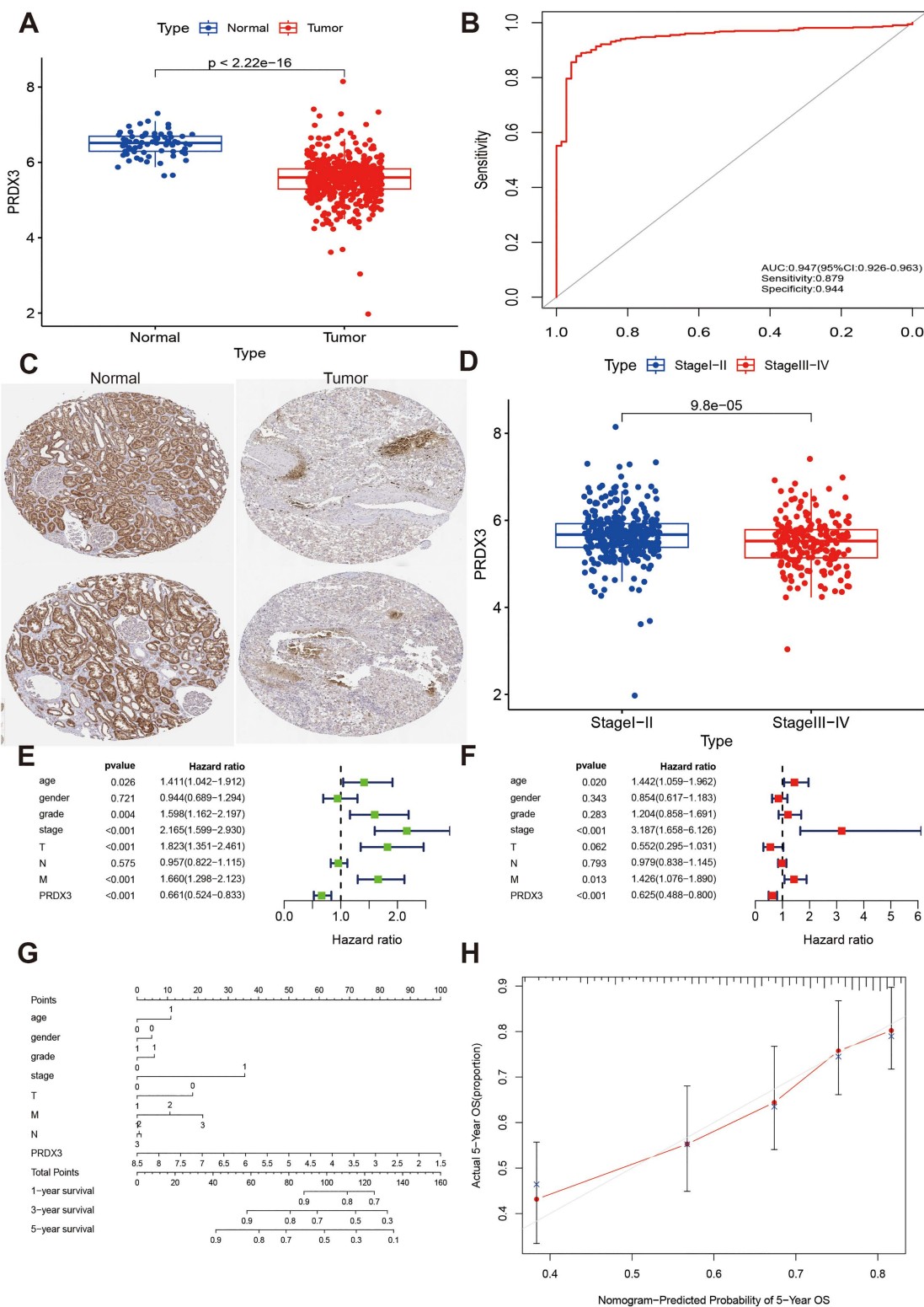

**Fig 8. PRDX3 independently predicted prognosis of KIRC patients. A:** low expression of PRDX3 in KIRC. **B:** ROC curve of PRDX3 in diagnosing KIRC. **C:** Immunohistochemistry of PRDX3 in KIRC and normal tissues. D: Association between PRDX3 and clinical stages. **E and F:** Univariate and multivariate cox regressions of PRDX3 in KIRC. **G:** Nomograph of PRDX3 predicting prognosis in KIRC. **H:** Calibration plot assessing the nomogram prediction model.

prognostic relevance across multiple clinically meaningful endpoints (OS, PFI, and DSS). Importantly, PRDX3 remained a significant factor in multivariable models in KIRC, supporting its potential independent prognostic value. We selected KIRC for experimental validation. We established PRDX3 over-expressing cells and control cells in the 786−0 and Caki-1 cell lines. Western blot and qPCR showed PRDX3 was over-expressed in 786−0 and Caki-1 cells (Figs 9A and 9B, S1 File). The scratch assay demonstrated PRDX3 over-expression inhibited the wound healing rate in KIRC cells (Fig 9C). Similarly, the migration and invasion abilities were also significantly inhibited in PRDX3 over-expressing cells (Figs 9D and 9E). These results suggest PRDX3 is a potential tumor suppressor gene.

To explore potential downstream mechanisms, KEGG enrichment analysis indicated that PRDX3 was associated with peroxisome and PPAR signaling (Fig. 10A). PRDX3 expression was positively correlated with PPARA (r = 0.346, P < 0.001; Fig 10B). In PRDX3-overexpressing KIRC cells, PPARA protein and mRNA levels were increased, as shown by Western blotting and RT–qPCR (Figs. 10C–10D; S1 File). qPCR also showed elevated PPARA expression level in PRDX3 over-expressing cells (Fig 10D). The PPAR signaling pathway was closely associated with inflammation and involved cell growth, proliferation, differentiation, metabolism, and microenvironment remodeling. We further detected some common genes that were related to cell growth, apoptosis, a microenvironment remodeling, including CCND, CCL20, MMP9, Bax, BMF, and Bim. Our results suggest that PRDX3 over-expression promotes the expressions of aptosis-promoting genes (Bax, BMF, and Bim) and inhibits the expressions of genes that are related to growth and invasion (CCND, CCL20, MMP9; Figs 10E and 10F). At last, we added the PPARA inhibitor in the PRDX3 over-expressing cells, and found the growth, migration and invasion abilities were significantly suppressed in KIRC cells (Figs 10G-10I and S3 Fig). These results indicate PRDX3 plays an antioncogene role via the PPAR signaling pathway.

## Discussion

Malignant tumors remain among the leading causes of death worldwide. Accordingly, elucidating tumor pathogenesis and identifying effective therapeutic targets and biomarkers are essential for improving early detection, optimizing treatment strategies, and ultimately enhancing long-term outcomes [26]. PRDX3 is a common REDOX protease involved in various physiological and pathological processes [27,28]. Increasing evidence suggests that PRDX3 contributes to the initiation and progression of multiple malignancies and that its expression is associated with patient prognosis, although its roles appear to be context dependent [29–31].

In this study, we first characterized PRDX3 expression in normal human tissues and observed broad expression with low tissue specificity. PRDX3 showed relatively higher expression in the liver, tongue, kidney, skeletal muscle, and adrenal gland, and comparatively lower expression in skin. We then analyzed PRDX3 expression across tumor and adjacent nor-mal tissues in TCGA and found clear tumor-type heterogeneity: PRDX3 was downregulated in CHOL, KICH, KIRC, and SARC, but upregulated in COAD, ESCA, GBM, LUAD, LUSC, PRAD, STAD, and UCEC. These findings support a tissue- and context-specific pattern of PRDX3 dysregulation. We further examined genomic and epigenetic correlates of PRDX3 expression. Overall, PRDX3 mutations were uncommon (generally <3% across most tumor types), and PRDX3 expres-sion showed limited association with mutation status; notably, only CESC exhibited lower PRDX3 expression in mutant versus wild-type cases. PRDX3 mutation frequency exceeded 3% only in miscellaneous neuroepithelial tumors, which may reflect the high mutational burden observed in neuroepithelial malignancies [32]. In contrast, PRDX3 copy-number variation was more frequent in several cancers (e.g., glioma, pleural mesothelioma, BLCA, SKCM, PRAD, and CESC), and PRDX3 expression generally increased with rising copy number. In addition, PRDX3 methylation was inversely cor-related with mRNA expression, suggesting epigenetic repression. Collectively, these results indicate that PRDX3 dysreg-ulation in cancer is shaped by multiple mechanisms, including CNV and DNA methylation. Consistent with this context dependence, survival analyses revealed bidirectional prognostic associations across tumor types. PRDX3 acted as a protective factor in BRCA, COAD, GBM, KIRC, KIRP, LGG, LIHC, LUSC, MESO, SARC, and SKCM, but as a risk factor in CESC, CHOL, DLBC, ESCA, HNSC, KICH, LAML, LUAD, PAAD, PRAD, and READ. These findings underscore the dual

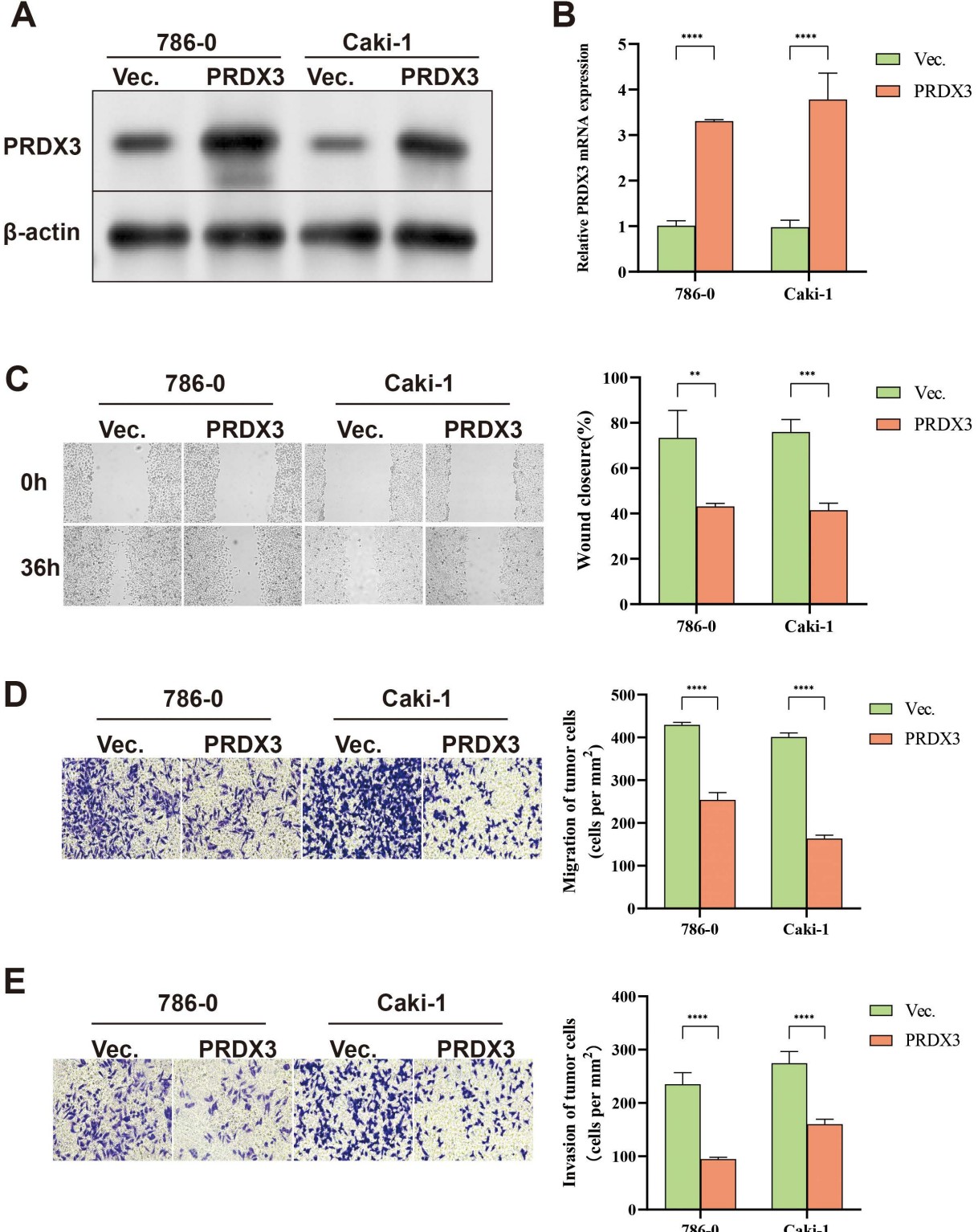

**Fig 9. PRDX3 over-expression inhibited tumor progression in KIRC cells. A:** Western blot showing PRDX3 over-expression in 786−0 and Caki-1 cells. **B:** qPCR showing elevated PRDX3 in KIRC cells. Data (n = 3) are shown as mean ± standard error (SE), * = P < 0.05; ** = P < 0.01; *** = P < 0.001;

\*\*\*\*=P<0.0001. **C:** PRDX3 over-expression inhibited wound healing. Data are shown as mean±standard error (SE), \*=P<0.05; \*\*=P<0.01; \*\*\*=P<0.001; \*\*\*\*=P<0.0001. **D and E:** PRDX3 over-expression inhibited migration and invasion in KIRC cells. Data are shown as mean±standard error (SE), \*=P<0.05; \*\*=P<0.01; \*\*\*=P<0.001; \*\*\*\*=P<0.0001.

roles of PRDX3 in tumor biology, likely reflecting differences in tumor lineage, metabolic dependencies, microenvironmental context, and co-regulated signaling programs..

Mechanistically, PRDX3 is a key component of mitochondrial redox homeostasis, and our pathway analyses were consistent with this role. PRDX3 expression was associated with oxidative stress–related and metabolic programs across cancers, including reactive oxygen species signaling, peroxisome, oxidative phosphorylation, and hypoxia. Cancer cells often exhibit elevated antioxidant capacity to buffer increased oxidative stress, and mitochondrial PRDXs have been reported to be upregulated in certain malignant contexts [33]. In addition to the high sensitivity to oxidative stress, PRDX3 is regulated by three classes of histone deacetylases (SIRT1) through enhancing the formation of the PGC-1α/FoxO3a transcription complex [34,35]. Different cancers and partners interact differently with PRDX3, resulting in different signaling and regulation. Several studies show PRDX3 is upregulated in cancer-associated cells. The question remains whether these PRDXs promote or inhibit cancers [36]. Studies show that PRDX3 and CD133 expressions are regulated by FOXM1 through binding to the promoters of PRDX3 and CD133. PRDX3 knockout mice showed reduced tumor volume and metastasis, providing clues to the association of PRDX3 with FoxM1-related cancer development pathways. To develop highly effective drugs, it is possible to target FOXM1 and FOXM1/PRDX3, which play a role in cancer cell proliferation and cell survival [36]. In addition, PRDX3 has been implicated in regulated cell death, including ferroptosis. In our analysis, PRDX3 expression was positively associated with ferroptosis- and cuproptosis-related pathway signatures across multiple tumor types, further supporting a link between PRDX3 and cell-death regulation..

We also evaluated PRDX3 in relation to the tumor microenvironment and immune status. PRDX3 correlated positively with TMB in CHOL, COAD, LUAD, LUSC, PRAD, READ, SKCM, STAD, TGCT, UCEC, and UCS, but negatively in BRCA and LGG, suggesting tumor-type–dependent associations with genomic immunogenicity. PRDX3 was negatively associated with immune scores in most cancers (with exceptions such as LUSC), and it showed broad correlations with immune infiltration signatures. Notably, PD-L1 (CD274) was positively correlated with PRDX3 in KIRP, DLBC, UVM, HNSC, SKCM, PAAD, and KIRC. Together, these findings suggest that PRDX3 may reflect, or potentially contribute to, microenvironmental states that influence immune infiltration and checkpoint signaling, and may therefore be relevant to immunotherapy responsiveness in certain contexts.

Finally, we validated the biological roles of PRDX3 in KIRC using extra experiments. Clinical data showed PRDX3 was independently associated with prognosis in KIRC and can be considered as a prognosis predictor. We built PRDX3 over-expression KIRC cells and found PRDX3 over-expression inhibited the progression of KIRC cells. KEEG pathway analysis showed PRDX3 was positively associated with PPAR signaling pathway, which is mediated by ligand-activated nuclear receptors (PPARα, PPARβ/δ, and PPARγ) that heterodimerize with RXR and bind PPAR response elements to regulate transcriptional programs controlling fatty-acid uptake/transport, lipid storage, and mitochondrial/peroxisomal fatty-acid oxidation, with additional roles in inflammation-related signaling. [37–39]. Our results also found that PRDX3 overexpression was associated with upregulation of pro-apoptotic genes (Bax, BMF, Bim) and downregulation of cell-cycle/invasion-related genes (CCND1, CCL20, MMP9). Thus, the observed reduction in migration/invasion may be driven by both direct effects on invasive programs and indirect effects due to altered proliferation/viability. Furthermore, we identified that PRDX3 affected PPARA, which is one of PPAR subunits. Inhibitors of PPARA had been developed to treat different diseases. The migration, invasion and wound healing abilities were significantly suppressed after adding PPARA inhibitors. Our results suggest PRDX3 may affect KIRC progression via the PPAR signaling pathway. From a translational and drug-development standpoint [40], this result may support PRDX3 as a plausible stratification biomarker

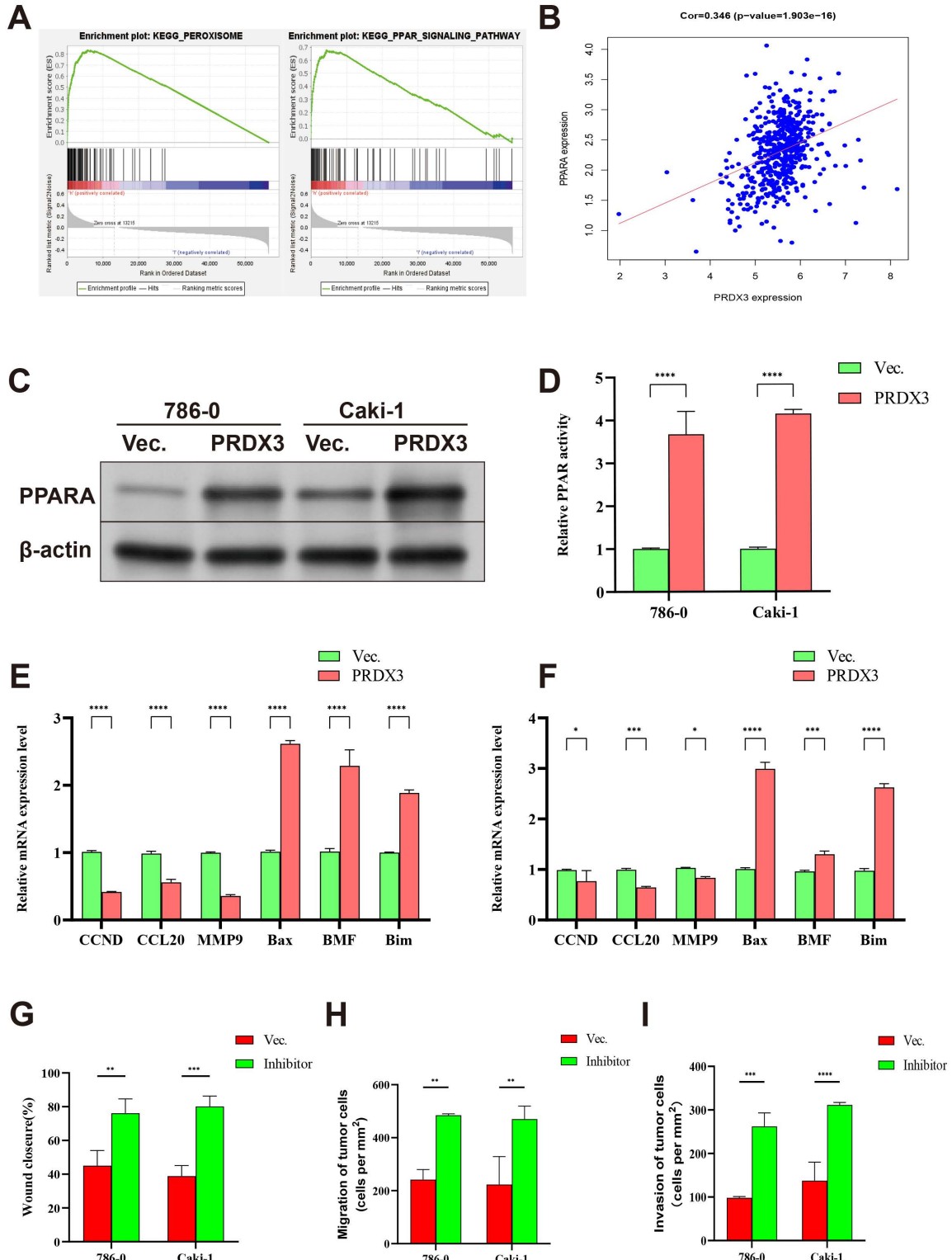

**Fig 10. PRDX3 inhibited KIRC progression via regulating PPAR signaling pathway. A:** KEGG pathway enrichment of PRDX3. **B:** Scatter plot of correlation of PRDX3 with PPARA. **C and D:** Western blot and qPCR showing high PPARA expression in PRDX3 over-expressing cells. The qPCR was shown using (n = 3) mean ± standard error (SE), * = P < 0.05; ** = P < 0.01; *** = P < 0.001; **** = P < 0.0001. **E and F:** PRDX3 promoting Bax, BMF, and

Bim expressions and inhibiting CCND, CCL20 and MMP9 expressions. Data (n = 3) are shown as mean ± standard error (SE), * = P < 0.05; ** = P < 0.01; *** = P < 0.001; **** = P < 0.0001. **G, H and I:** wound healing, migration and invasion abilities suppressed after adding PPAR inhibitor. Data (n = 3) are shown as mean ± standard error (SE), * = P < 0.05; ** = P < 0.01; *** = P < 0.001; **** = P < 0.0001.

in ccRCC, and our enrichment results implicate metabolic programs including the PPAR signaling axis. PPAR activity has been linked to metabolic reprogramming, antioxidant defenses, and immunosuppressive microenvironmental features that may contribute to therapeutic resistance; therefore, PRDX3–PPAR co-activation patterns raise a testable hypothesis that PRDX3 could help identify subsets of patients more reliant on lipid-metabolic/redox pathways and potentially inform pathway-guided targeting or rational combinations.

Our study has several limitations. First, our TCGA-based analyses follow an established and widely adopted strategy in cancer bioinformatics, and Similar TCGA-driven frameworks have been applied to dissect oncogenic signaling programs in oral squamous cell carcinoma and to construct ssGSEA-based immune prognostic models in glioblastoma, supporting the methodological tradition underlying our approach [41,42]. But bulk transcriptomic resources are not neutral truth sources and inherently contain both technical and biological biases [43,44]. Therefore, our TCGA-based results should be interpreted as associative and hypothesis-generating rather than causal, and key observations require validation in independent cohorts and orthogonal experimental systems. Second, the clinical samples were from public data, and association between PRDX3 and prognosis in KIRC needs to be validated in an extra dataset. Third, we did not directly quantify intracellular or mitochondrial ROS following PRDX3 manipulation; therefore, the redox-mediated interpretation remains inferential and requires confirmation with ROS assays and rescue experiments in future studies Finally, previous studies reported PRDX3 was associated with ferroptosis, but this association was not found in KIRC. We only validated our findings on one type of cancer, and validations shall be performed in multiple types of cancer.

In conclusion, PRDX3 gene is generally differentially expressed in human malignant tumors, and its expression level may be regulated by gene copy number variation and methylation level. The expression level of PRDX3 gene is closely related to the prognosis of patients and plays dual roles. PRDX3 may regulate tumor progression by affecting tumor cell division, metabolism, and REDOX processes in the body. In vitro experiments show that PRDX3 can inhibit the progression of KIRC cells by linking the PPAR pathway. Cancer therapy is increasingly moving toward precision strategies that integrate tumor-intrinsic vulnerabilities with the immune microenvironment [45], and our findings nominate PRDX3 as a hypothesis-generating marker of redox–metabolic state (including PPAR-related programs) that could eventually support patient stratification and rational combination concepts. These translational implications remain preliminary and will require validation in treatment-annotated cohorts plus mechanistic and spatial/single-cell studies to confirm actionable PRDX3-linked dependencies.

## Supporting information

**S1 Table. Lists of cancer types of abbreviations.**
(DOCX)

**S2 Table. The list of primers used in the reactions for reverse transcription qPCR.**
(DOCX)

**S1 Fig. Association between PRDX3 expression and methylation level in cancers.**
(TIF)

**S2 Fig. Association between PRDX3 and stromal score in cancers.**
(TIF)

**S3 Fig. PRDX3-metiated tumor suppression via the PPAR signaling pathway in KIRC.**
(TIF)

**S1 File. Original uncropped and unadjusted images underlying all blot results.**
(PDF)

## Author contributions

**Conceptualization:** Peng Chen.

**Data curation:** Yanyan Li, Xiangyuan Tan, Zhanzhan Li.

**Formal analysis:** Zhanzhan Li.

**Funding acquisition:** Peng Chen.

**Investigation:** Xiangyuan Tan, Zhanzhan Li.

**Methodology:** Zhanzhan Li.

**Resources:** Xiangyuan Tan.

**Software:** Zhanzhan Li.

**Supervision:** Peng Chen.

**Validation:** Yanyan Li, Xiangyuan Tan.

**Visualization:** Zhanzhan Li, Peng Chen.

**Writing – original draft:** Yanyan Li.

**Writing – review & editing:** Xiangyuan Tan, Zhanzhan Li, Peng Chen.

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
