## [Decision Letter · Decision Letter 0]

19 Jan 2026

Dear Dr. Chen,

Thank you for submitting your manuscript to PLOS ONE. After careful consideration, we feel that it has merit but does not fully meet PLOS ONE’s publication criteria as it currently stands. Therefore, we invite you to submit a revised version of the manuscript that addresses the points raised during the review process.

We look forward to receiving your revised manuscript.

Kind regards,

Xiaoen Wei

Academic Editor

PLOS One

Journal Requirements:

3. We note that Figure 1 in your submission contain copyrighted images. All PLOS content is published under the Creative Commons Attribution License (CC BY 4.0), which means that the manuscript, images, and Supporting Information files will be freely available online, and any third party is permitted to access, download, copy, distribute, and use these materials in any way, even commercially, with proper attribution. For more information, see our copyright guidelines: http://journals.plos.org/plosone/s/licenses-and-copyright.

Reviewers' comments:

Reviewer's Responses to Questions

**Comments to the Author**

1. Is the manuscript technically sound, and do the data support the conclusions?

Reviewer #1: Yes

Reviewer #2: Partly

Reviewer #3: Yes

2. Has the statistical analysis been performed appropriately and rigorously?

Reviewer #1: Yes

Reviewer #2: Yes

Reviewer #3: Yes

3. Have the authors made all data underlying the findings in their manuscript fully available?

Reviewer #1: Yes

Reviewer #2: Yes

Reviewer #3: Yes

4. Is the manuscript presented in an intelligible fashion and written in standard English?

Reviewer #1: Yes

Reviewer #2: Yes

Reviewer #3: Yes

Reviewer #1: This is an interesting pan-cancer plus mechanism paper with real experimental effort, but it still feel overextended and under-contextualized in places, so a major revision make sense to tighten the logic and strengthen the translational angle.

In the introduction, the background read solid but also a bit generic, and it would really benefit from stepping back and framing PRDX3 within the broader cancer biomarker landscape instead of jumping so fast into oxidative stress biology. Biomarker discovery is not just about association but about eventual utility, and this is where a short conceptual bridge help a lot. I strongly suggest anchoring the rationale by briefly discussing how discovery-stage biomarkers move toward prognosis, stratification, and eventually intervention, for example by citing “Bridging Discovery and Treatment: Cancer Biomarker,” which fit naturally here and give the reader a clearer sense of why PRDX3 matter beyond being another differentially expressed gene. That kind of framing would also make the pan-cancer scope feel more intentional rather than simply exhaustive.

For the results sections that rely heavily on TCGA analyses, the overall workflow is acceptable and quite standard, but the manuscript treat TCGA almost as a neutral truth source, which is risky. Many readers already know this, but it still need to be said explicitly. It would help to situate the approach within prior TCGA-based efforts, such as TCGA-Based Analysis of Oncogenic Signaling Pathways Underlying Oral Squamous Cell Carcinoma or Immune prognostic model for glioblastoma based on the ssGSEA enrichment score, just to show that the strategy follow an established analytical tradition. At the same time, this is exactly where a more critical tone is needed. Bulk transcriptomic resources come with built-in biases, in sample composition, stromal contamination, batch effects, and clinical heterogeneity, and ignoring this weaken the interpretation of differential expression, immune infiltration, and pathway enrichment. A brief but explicit acknowledgement, supported by “Genetic Expression in Cancer Research: Challenges and Complexity” (2024) and “Technical and Biological Biases in Bulk Transcriptomic Data Mining for Cancer Research” (2025), would go a long way toward making the conclusions feel more measured and trustworthy.

When moving into the functional and immune-related analyses, the story become interesting but also a bit scattered. The immune microenvironment results are extensive, yet the biological implications are not always clear, and the paper sometimes describe correlation without really slowing down to explain what that might mean in a tumor context. I think the authors should lean more into interpretation here, especially where PRDX3 appear linked to immune modulation, checkpoint markers, or stromal features. Even a few sentences per subsection that ask what this could imply for therapy response or resistance would make the data feel less like a catalogue and more like a hypothesis-generating framework. here you should mention review paper "Identification of new immune target and signaling for cancer immunotherapy".

The experimental validation in KIRC is a strength, but the discussion stop short of translating this into a broader drug development perspective. Biomarkers do not live in isolation, and the manuscript would benefit from explicitly addressing how PRDX3-associated pathways, particularly the PPAR axis, could inform therapeutic targeting, patient selection, or combination strategies. In this context, citing and briefly discussing “Fast, tracking drug development with biomarkers and companion diagnostics.” would fit very naturally and help bridge the gap between molecular mechanism and clinical application. This does not require overclaiming, just a clearer statement of how such a biomarker could realistically be used.

Finally, the discussion and future outlook feel a bit conservative given how much data is presented. With deeper insight into tumor biology, cancer treatment is clearly moving toward personalized and precision approaches, especially those targeting the tumor immune microenvironment and specific molecular dependencies. I encourage the authors to end on a more forward-looking note, connecting their findings to emerging therapeutic paradigms and the idea that PRDX3-related vulnerabilities could eventually support tailored intervention strategies. Referencing “Overview of perspectives on cancer, newer therapies, and future directions” would help frame this vision and signal that the work is aligned with where the field is heading, even if much validation still remain to be done.

Reviewer #2: This study presents an interesting multi-omics investigation into the role of PRDX3 across 33 cancer types, with a specific focus on Kidney Renal Clear Cell Carcinoma (KIRC). While the bioinformatic integration is a clear strength, the manuscript requires additional experimental depth and technical clarification to support the authors’ mechanistic claims.

- Please add recent statistics regarding kidney cancer prevalence and survival rates in the Introduction to provide a clearer background on the disease’s impact.

- Correlation between PRDX3 expression and PPAR pathway markers via Western Blot is insufficient to claim a regulatory relationship.

- Clarify if the reduced migration/invasion observed in KIRC cells is a direct effect on motility or a secondary effect of decreased cell proliferation.

- The study notes that PRDX3 exhibits "dual roles" in tumor biology across different cancers. Please provide a table or supplementary figure clarifying which of the 33 cancers show PRDX3 as an oncogene/ an antioncogene.

- The authors should provide data on the intracellular ROS levels in KIRC cell models after PRDX3 manipulation.

- Please justify the choice of KIRC cell lines used. Are these lines representative of the "high-methylation" or "low-methylation" groups identified in your TCGA analysis?

- While the in vitro data supports the role of PRDX3 in KIRC, the lack of animal models makes it difficult to assess the tumor microenvironment and immune response mentioned in the conclusion.

- The authors state that 50 pg of protein was loaded for Western blot analysis. Standard protocols typically employ 20 - 50 µg. Could the authors clarify if this is a typographical error? If 50 pg was indeed the amount used, please provide details on the high-sensitivity detection system or ultra-pure recombinant protein used to achieve a signal at this level.

- The English needs improvement, and there are some errors.

- Be careful of abbreviations.

Reviewer #3: 1. Why do authors particularly chose KIRC not any other cancer type cell lines that are more significant in tumor microenvirnoment? Justify.

2. How the findings can be clinically applied?

3. Fig.2 represent the same information so except Fig.2A others can be moved to supplement file.

4. Clarify the term TBM whether it is TMB or TBM

5. Fig. 4, Fig. 6, Fig.7 results does have much correlation with KIRC. Justify their importance or move them to supplement file.

6. Give a brief description of peroxisome proliferator-activated receptor signaling pathway in KIRC.

7. Give a pictorial representation of the identified pathway mechanism.

**Do you want your identity to be public for this peer review?** For information about this choice, including consent withdrawal, please see our Privacy Policy

Reviewer #1: **Yes:** Yingcai Hong

Reviewer #2: No

Reviewer #3: **Yes:** Priyadharshini Annadurai

---

## [Author Response · Author response to Decision Letter 1]

4 Feb 2026

Response to Editor and Reviewers

Response to Editorial comments

Journal Requirements: When submitting your revision, we need you to address these additional requirements.

Comment 1: Please ensure that your manuscript meets PLOS ONE's style requirements, including those for file naming. The PLOS ONE style templates can be found at https://journals.plos.org/plosone/s/file?id=wjVg/PLOSOne_formatting_sample_main_body.pdf and https://journals.plos.org/plosone/s/file?id=ba62/PLOSOne_formatting_sample_title_authors_affiliations.pdf

Response 1: Yes, we have revised our manuscript according to these two samples, which meets PLOS ONE's style requirements.

Comment 2: PLOS requires an ORCID iD for the corresponding author in Editorial Manager on papers submitted after December 6th, 2016. Please ensure that you have an ORCID iD and that it is validated in Editorial Manager. To do this, go to ‘Update my Information’ (in the upper left-hand corner of the main menu), and click on the Fetch/Validate link next to the ORCID field. This will take you to the ORCID site and allow you to create a new iD or authenticate a pre-existing iD in Editorial Manager.

Comment 2: Yes, we have provided the ORCID iD for the corresponding author.

Comment 3: We note that Figure 1 in your submission contains copyrighted images. All PLOS content is published under the Creative Commons Attribution License (CC BY 4.0), which means that the manuscript, images, and Supporting Information files will be freely available online, and any third party is permitted to access, download, copy, distribute, and use these materials in any way, even commercially, with proper attribution. For more information, see our copyright guidelines: http://journals.plos.org/plosone/s/licenses-and-copyright. We require you to either (1) present written permission from the copyright holder to publish these figures specifically under the CC BY 4.0 license, or (2) remove the figures from your submission:1. You may seek permission from the original copyright holder of Figure 1 to publish the content specifically under the CC BY 4.0 license. We recommend that you contact the original copyright holder with the Content Permission Form (http://journals.plos.org/plosone/s/file?id=7c09/content-permission-form.pdf) and the following text: “I request permission for the open-access journal PLOS ONE to publish XXX under the Creative Commons Attribution License (CCAL) CC BY 4.0 (http://creativecommons.org/licenses/by/4.0/). Please be aware that this license allows unrestricted use and distribution, even commercially, by third parties. Please reply and provide explicit written permission to publish XXX under a CC BY license and complete the attached form.” Please upload the completed Content Permission Form or other proof of granted permissions as an "Other" file with your submission. In the figure caption of the copyrighted figure, please include the following text: “Reprinted from [ref] under a CC BY license, with permission from [name of publisher], original copyright [original copyright year].” 2. If you are unable to obtain permission from the original copyright holder to publish these figures under the CC BY 4.0 license or if the copyright holder’s requirements are incompatible with the CC BY 4.0 license, please either i) remove the figure or ii) supply a replacement figure that complies with the CC BY 4.0 license. Please check copyright information on all replacement figures and update the figure caption with source information. If applicable, please specify in the figure caption text when a figure is similar but not identical to the original image and is therefore for illustrative purposes only.

Response 3: Thank you. We have removed the copyrighted images and used our own figures in Figure 1.

Comment 4: PLOS ONE now requires that authors provide the original uncropped and unadjusted images underlying all blot or gel results reported in a submission’s figures or Supporting Information files. This policy and the journal’s other requirements for blot/gel reporting and figure preparation are described in detail at https://journals.plos.org/plosone/s/figures#loc-blot-and-gel-reporting-requirements and https://journals.plos.org/plosone/s/figures#loc-preparing-figures-from-image-files. When you submit your revised manuscript, please ensure that your figures adhere fully to these guidelines and provide the original underlying images for all blot or gel data reported in your submission. See the following link for instructions on providing the original image data: https://journals.plos.org/plosone/s/figures#loc-original-images-for-blots-and-gels. In your cover letter, please note whether your blot/gel image data are in Supporting Information or posted at a public data repository, provide the repository URL if relevant, and provide specific details as to which raw blot/gel images, if any, are not available. Email us at plosone@plos.org if you have any questions.

Response 4: Yes, we have provided original uncropped and unadjusted images underlying all blot or gel results reported in a submission’s figures and uploaded these images in the Supporting Information files (S1_File.pdf).

Comment 5: If the reviewer’s comments include a recommendation to cite specific previously published works, please review and evaluate these publications to determine whether they are relevant and should be cited. There is no requirement to cite these works unless the editor has indicated otherwise.

Response 5: Yes, we have review and evaluate these publications very carefully and make a cautious citation.

Response to Reviewers

Response to Reviewer #1:

This is an interesting pan-cancer plus mechanism paper with real experimental effort, but it still feel overextended and under-contextualized in places, so a major revision make sense to tighten the logic and strengthen the translational angle.

Comment 1: In the introduction, the background read solid but also a bit generic, and it would really benefit from stepping back and framing PRDX3 within the broader cancer biomarker landscape instead of jumping so fast into oxidative stress biology. Biomarker discovery is not just about association but about eventual utility, and this is where a short conceptual bridge help a lot. I strongly suggest anchoring the rationale by briefly discussing how discovery-stage biomarkers move toward prognosis, stratification, and eventually intervention, for example by citing “Bridging Discovery and Treatment: Cancer Biomarker,” which fit naturally here and give the reader a clearer sense of why PRDX3 matter beyond being another differentially expressed gene. That kind of framing would also make the pan-cancer scope feel more intentional rather than simply exhaustive.

Response 1: Thank you for your advice. We added the following descriptions in the introduction as follows: Therefore, clarifying tumor etiology and identifying clinically useful biomarkers and therapeutic targets remain critical for earlier detection, improved risk stratification, and better long-term outcomes. Importantly, biomarker research is not only about statistical association but also about eventual clinical utility. A biomarker is broadly defined as a measurable characteristic that reflects normal or pathogenic processes or responses to an intervention, and cancer biomarkers may serve diagnostic, prognostic, predictive, or monitoring roles depending on their intended “context of use”[6]. In general, discovery-stage candidates should be evaluated along a translational continuum that includes analytical validation (reliable and reproducible measurement), clinical validation (replication of associations with relevant clinical outcomes), and finally clinical utility (demonstrated benefit for prognosis, stratification, or treatment decision-making). Within this framework, pan-cancer analyses can be particularly informative at the discovery stage by assessing the robustness and tumor-type specificity of biomarker signals and by prioritizing cancer types for focused validation and mechanistic studies.

6. Liu H, Karsidag I, Golin R, Wu G. Bridging Discovery and Treatment: Cancer Biomarker. Cancers. 2025;17(22):3720. http://doi.org/10.3390/cancers17223720

Comment 2: For the results sections that rely heavily on TCGA analyses, the overall workflow is acceptable and quite standard, but the manuscript treat TCGA almost as a neutral truth source, which is risky. Many readers already know this, but it still need to be said explicitly. It would help to situate the approach within prior TCGA-based efforts, such as TCGA-Based Analysis of Oncogenic Signaling Pathways Underlying Oral Squamous Cell Carcinoma or Immune prognostic model for glioblastoma based on the ssGSEA enrichment score, just to show that the strategy follow an established analytical tradition. At the same time, this is exactly where a more critical tone is needed. Bulk transcriptomic resources come with built-in biases, in sample composition, stromal contamination, batch effects, and clinical heterogeneity, and ignoring this weaken the interpretation of differential expression, immune infiltration, and pathway enrichment. A brief but explicit acknowledgement, supported by “Genetic Expression in Cancer Research: Challenges and Complexity” (2024) and “Technical and Biological Biases in Bulk Transcriptomic Data Mining for Cancer Research” (2025), would go a long way toward making the conclusions feel more measured and trustworthy.

Response 2: Thank you for your advice. We added the following descriptions in study limitations section and cited these studies: First, our TCGA-based analyses follow an established and widely adopted strategy in cancer bioinformatics, and Similar TCGA-driven frameworks have been applied to dissect oncogenic signaling programs in oral squamous cell carcinoma and to construct ssGSEA-based immune prognostic models in glioblastoma, supporting the methodological tradition underlying our approach[41, 42]. But bulk transcriptomic resources are not neutral truth sources and inherently contain both technical and biological biases[43, 44]. Therefore, our TCGA-based results should be interpreted as associative and hypothesis-generating rather than causal, and key observations require validation in independent cohorts and orthogonal experimental systems.

41. Gao X, Liu D, Liu J, Wang R. TCGA-based analysis of oncogenic signaling pathways underlying oral squamous cell carcinoma. Oncology and Translational Medicine. 2024;10(2):87-92. http://doi.org/10.1097/ot9.0000000000000025

42. Okamoto T, Mizuta R, Demachi-Okamura A, Muraoka D, Sasaki E, Masago K, et al. Immune prognostic model for glioblastoma based on the ssGSEA enrichment score. Cancer Genet-Ny. 2025;294-295:32-41. http://doi.org/10.1016/j.cancergen.2025.03.005

43. Liu H, Li Y, Karsidag M, Tu T, Wang P. Technical and Biological Biases in Bulk Transcriptomic Data Mining for Cancer Research. J Cancer. 2025;16(1):34-43. http://doi.org/10.7150/jca.100922

44. Liu H, Guo Z, Wang P. Genetic expression in cancer research: Challenges and complexity. Gene Reports. 2024;37:102042. http://doi.org/10.1016/j.genrep.2024.102042

Comment 3: When moving into the functional and immune-related analyses, the story become interesting but also a bit scattered. The immune microenvironment results are extensive, yet the biological implications are not always clear, and the paper sometimes describe correlation without really slowing down to explain what that might mean in a tumor context. I think the authors should lean more into interpretation here, especially where PRDX3 appear linked to immune modulation, checkpoint markers, or stromal features. Even a few sentences per subsection that ask what this could imply for therapy response or resistance would make the data feel less like a catalogue and more like a hypothesis-generating framework. here you should mention review paper "Identification of new immune target and signaling for cancer immunotherapy".

Response 3: Thank you for your advice. We agree that the functional and immune-related results, while comprehensive, could be read as a catalogue of correlations without sufficient biological interpretation in tumor context. In the revised manuscript, we reorganized the immune-related Results into a clearer narrative (TMB/MSI → ESTIMATE immune/stromal scores, immune cell infiltration, immune checkpoint/co-stimulatory genes, chemokines/receptors) and added brief interpretive sentences at the end of each subsection. Specifically, we now discuss what PRDX3-associated immune patterns could imply for immune activation vs. immune suppression/exclusion, potential adaptive immune resistance (checkpoint upregulation), and plausible links to immunotherapy response or resistance, while keeping the conclusions appropriately hypothesis-generating. We also cited the study.

24. Narote S, Desai SA, Patel VP, Deshmukh R, Raut N, Dapse S. Identification of new immune target and signaling for cancer immunotherapy. Cancer Genet-Ny. 2025;294-295:57-75. http://doi.org/10.1016/j.cancergen.2025.03.004

Comment 4: The experimental validation in KIRC is a strength, but the discussion stop short of translating this into a broader drug development perspective. Biomarkers do not live in isolation, and the manuscript would benefit from explicitly addressing how PRDX3-associated pathways, particularly the PPAR axis, could inform therapeutic targeting, patient selection, or combination strategies. In this context, citing and briefly discussing “Fast, tracking drug development with biomarkers and companion diagnostics.” would fit very naturally and help bridge the gap between molecular mechanism and clinical application. This does not require overclaiming, just a clearer statement of how such a biomarker could realistically be used.

Response 4: Thank you for your advice. We added the following descriptions in the discussion: From a translational and drug-development standpoint[40], this result may support PRDX3 as a plausible stratification biomarker in ccRCC, and our enrichment results implicate metabolic programs including the PPAR signaling axis. PPAR activity has been linked to metabolic reprogramming, antioxidant defenses, and immunosuppressive microenvironmental features that may contribute to therapeutic resistance; therefore, PRDX3–PPAR co-activation patterns raise a testable hypothesis that PRDX3 could help identify subsets of patients more reliant on lipid-metabolic/redox pathways and potentially inform pathway-guided targeting or rational combinations.

40. McBrearty N, Bahal D, Platero S. Fast-tracking drug development with biomarkers and companion diagnostics. Journal of Cancer Metastasis and Treatment. 2024;10:3. http://doi.org/10.20517/2394-4722.2023.134

Comment 5: Finally, the discussion and future outlook feel a bit conservative given how much data is presented. With deeper insight into tumor biology, cancer treatment is clearly moving toward personalized and precision approaches, especially those targeting the tumor immune microenvironment and specific molecular dependencies. I encourage the authors to end on a more forward-looking note, connecting their findings to emerging therapeutic paradigms and the idea that PRDX3-related vulnerabilities could eventually support tailored intervention strategies. Referencing “Overview of perspectives on cancer, newer therapies, and future directions” would help frame this vision and signal that the work is aligned with where the field is heading, even if much validation still remain to be done.

Response 5: Thank you for your advice. We revised the descriptions as follows: Cancer therapy is increasingly moving toward precision strategies that integrate tumor-intrinsic vulnerabilities with the immune microenvironment[45], and our findings nominate PRDX3 as a hypothesis-generating marker of redox–metabolic state (including PPAR-related programs) that could eventually support patient stratification and rational combination concepts. These translational implications remain preliminary and will require validation in treatment-annotated cohorts plus mechanistic and spatial/single-cell studies to confirm actionable PRDX3-linked dependencies.

45. Joshi RM, Telang B, Soni G, Khalife A. Overview of perspectives on cancer, newer therapies, and

---

## [Decision Letter · Decision Letter 1]

2 Mar 2026

Multiple omics analyses and experiments validation identify PRDX3 as a biomarker of prognosis and antioncogene in kidney clear cell carcinoma

PONE-D-25-58601R1

Dear Dr. Chen,

We’re pleased to inform you that your manuscript has been judged scientifically suitable for publication and will be formally accepted for publication once it meets all outstanding technical requirements.

Kind regards,

Xiaoen Wei

Academic Editor

PLOS One

Additional Editor Comments (optional):

The authors have satisfactorily addressed the major concerns raised in the previous round of review

Reviewers' comments:

Reviewer's Responses to Questions

**Comments to the Author**

Reviewer #2: (No Response)

2. Is the manuscript technically sound, and do the data support the conclusions?

Reviewer #2: Yes

3. Has the statistical analysis been performed appropriately and rigorously?

Reviewer #2: Yes

4. Have the authors made all data underlying the findings in their manuscript fully available?

Reviewer #2: Yes

5. Is the manuscript presented in an intelligible fashion and written in standard English?

Reviewer #2: Yes

Reviewer #2: (No Response)

**Do you want your identity to be public for this peer review?** For information about this choice, including consent withdrawal, please see our Privacy Policy

Reviewer #2: No

---

## [Editor Report · Acceptance letter]

PONE-D-25-58601R1

PLOS One

Dear Dr. Chen,

I'm pleased to inform you that your manuscript has been deemed suitable for publication in PLOS One. Congratulations! Your manuscript is now being handed over to our production team.

Kind regards,

on behalf of

Dr. Xiaoen Wei

Academic Editor

PLOS One